# Breaking the silence: Barriers to help-seeking among female victims of domestic violence in Ardabil, Iran – A qualitative study

Samaneh Dabagh Fekri[1,2], Negar Khoshnevis[3], Armin Zareiyan[4], Elham Kheirkhahi[5], Zahra Behboodi Moghadam[1]*, Masoumeh Namazi[1]

1 Department of Midwifery and Reproductive Health, School of Nursing and Midwifery, Tehran University of Medical Sciences, Tehran, Iran, 2 Department of Midwifery, School of Nursing and Midwifery, Ardabil University of Medical Sciences, Ardabil, Iran, 3 Department of Forensic Toxicology, Legal Medicine Research Center, Legal Medicine Organization, Ardabil, Iran, 4 Department of Public Health, School of Nursing, Aja University of Medical Sciences, Tehran, Iran, 5 Department of Education, Legal Medicine Research Center, Legal Medicine Organization, Ardabil, Iran

* behboodi@tums.ac.ir

## Abstract

### Background

Globally, 1 in 3 women (approximately 30%) experience physical or sexual violence in their lifetime, with intimate partner violence being the most common form Therefore, official support for women victims of domestic violence, as the most important individuals in achieving sustainable development goals, leads to improved health and empowerment of women, ultimately increasing their productivity in society. The present study aimed to elucidate understanding and experiences of Iranian victims of domestic violence regarding barriers to seeking help.

### Methods

The present qualitative study employed purposive sampling and in-depth semi-structured interviews with 20 women who were victims of domestic violence from July 2023 to March 2024 at the Legal Medicine Organization, Ardabil, Iran. Qualitative content analysis was conducted conventionally using the method proposed by Zhang and Wildemuth (2016) with MAXQDA software version 10, VERBI Software GmbH, Berlin.

### Results

The age range of the participants was 15–45 years old with an average age of 30.65 years and the educational level of them ranged from elementary school to postgraduate degrees. The educational backgrounds of their spouses varied from elementary school to doctorate degree. The data analysis yielded three main categories and twelve Subcategories. The categories included "structural and economic barriers to

**Data availability statement:** We acknowledge that qualitative interview data are 'data'. Due to ethical reasons, the full dataset cannot be shared as it contains potentially identifying and sensitive information about participants. The limitations are enforced by the Ethics Committee of Tehran University of Medical Sciences (ethics code: IR.TUMS.FNM. REC.1402.094). Data access requests may be directed to [fnm-researchdeputy@tums.ac.ir].

**Funding:** The author(s) received no specific funding for this work.

**Competing interests:** The authors have declared that no competing interests exist.

**Abbreviations:** DV, domestic violence; DALYs, disability-adjusted life years; IPV, intimate partner violence

empowerment", "ineffective support providers", "and efforts and struggles to preserve the family".

## Conclusion

This study findings underscore a challenging reality for women, indicating a lengthy journey to assert their rights. Societal support for victims of domestic violence is insufficient, which exacerbates the situation, leading to further exploitation by men and heightened levels of violence. Victims of domestic violence require immediate access to specialized services, financial and moral support from official agencies and facilitating the formal help-seeking process.

## Background

Domestic violence (DV) is the most common cause of intentional injuries in women [1–3]. Globally, 1 in 3 women (approximately 30%) experience physical or sexual violence in their lifetime, with intimate partner violence being the most common form [4]. In Iran, recent studies indicate that domestic violence affects up to 66% of women, with significant regional variations [1]. Ardabil province is located in the northwest of Iran in Azerbaijan region [5]. Domestic violence is one of the greatest challenges to women's health in this province. The amount of domestic violence in all areas of Ardabil province is 29.6 percent [6]. This experience is closely associated with an increase in medical issues such as physical injuries, mental illnesses, and fetal abortions [2,7].

Violence against women is a strong indicator of the erosion of moral standards, violating women's fundamental human rights and freedoms, and hindering the attainment of peace and equality. The absence of laws supporting women and the presence of laws that perpetuate such violence have entrenched the persistence of this phenomenon in Iranian society [8].

Most victims of domestic violence live in isolation [9]. However, at a certain stage of the abusive relationship, women tend to seek help as they come to realize that the accumulated problems are insurmountable, and the escalation of physical violence prompts an increased request for assistance [10]. The process of seeking help involves both informal (family, friends, relatives, neighbors) and formal (police, medical centers, domestic violence agencies, etc.) avenue [11]. A study by Bibi *et al*. (2014) indicated that while approximately half of the domestic violence victims sought help from their relatives, around 48% did not seek help likely due to fear, limited access to resources, or societal constraints rather than an explicit preference for silence [12]. Choden (2019) reported that even two-thirds of women experiencing physical violence never seek help [10]. These findings can be interpreted in terms of deficiencies in the legal system, such as the unpreparedness of law enforcers to deal with domestic violence cases, stereotypic beliefs and attitudes, as well as the lack of specific training [13]. The negative experience of seeking help from institutions, the hostile environment at home, and the lack of support for victims only perpetuate learned helplessness in domestic violence victims [10], leading to the belief that

resistance is futile [14]. This is because previous attempts to escape violence have been unsuccessful and have even led to increased suffering. On the other hand, due to the stigma prevalent in society, fear of blame, misunderstandings, and further harm by the perpetrator, victims of domestic violence avoid seeking help [15,16]. Despite the many consequences, a large part of Iranian women experiencing IPV do not seek help. Several factors may account for this, such as keeping marital conflict private [17–19]. Lack of professionalism, mistrust and not understanding the nature of formal support [17,18]. Other factors, such as economic dependence, low educational status, and unemployment are associated with staying with an abusive partner [17,20]. Staying with an abusive partner is sometimes facilitated by unhelpful responses from service providers, such as being dismissed, blamed, or advised to remain silent for the sake of family unity [21].

While numerous studies have explored domestic violence globally, there is limited qualitative research on the barriers to help-seeking among women in Iran, particularly in regions like Ardabil province. This study addresses this gap by providing an in-depth understanding of the cultural, social, and structural barriers faced by Iranian women, offering context-specific insights that are crucial for developing effective interventions.

### Study aim

This study aims to explore the barriers to help-seeking among female victims of domestic violence in Ardabil, Iran, with a focus on understanding the cultural, social, and structural factors that influence their decisions.

## Materials and methods

### Ethics statement

Ethical approval was obtained from Ethics Committee of Tehran University of Medical Sciences (ethics code: IR.TUMS. FNM.REC.1402.094). In addition to explaining the study's objectives and informing participants about the available support services for women victims, written informed consent was obtained from all participants and informed consent was obtained from the parents or legal guardians of all minor participants prior to their inclusion in the study. This process was conducted in accordance with the ethical guidelines approved by Ethics Committee of Tehran University of Medical Sciences. They were assured that they could withdraw from the study at any time if they wished. If any of the questions caused discomfort to the participants, the interview was paused and resumed after a while with the participant's permission. Participants' privacy and confidentiality of data were preserved. Each participant was assigned a code to maintain confidentiality of their information.

### Study design

The current study was conducted qualitatively to gain deeper insight into the understanding and experiences of Iranian domestic violence victims regarding the barriers to seeking help in the sensitive issue of domestic violence. Qualitative description follows the principles of qualitative research and is the chosen method in cases of direct description of phenomena.

This method is suitable for answering "who," "what," and "where" questions about behaviors, motivations, and human perspectives, and involves a logical set of strategies for conceptual orientation, sampling, data construction, analysis, and reporting, which public health researchers aim to use for a descriptive-interpretive approach to develop knowledge about human well-being and disease.

### Setting and participants

The present study was conducted at the Legal Medicine Organization in Ardabil from July 27, 2023, to March 20, 2024. Purposeful sampling was employed based on maximum variation in age (15–45 years), occupational status, education level, duration of marriage, number of children, and socio-economic status to select suitable participants (Table 1).

**Table 1. Profile of participants.**

| Participant | Age of woman | Age of husband | Education of woman | Education of husband | Woman's job | Husband's job | Marriage duration (Years) | Number of Children | Socioeconomic status |
|---|---|---|---|---|---|---|---|---|---|
| 1 | 31 | 34 | Secondary | Secondary | Carpet weaver | Driver | 15years | 2 | Low |
| 2 | 45 | 52 | High school diploma | Master's degree | Hosewife | Retired teacher | 29 | 3 | Moderate |
| 3 | 25 | 28 | High school diploma | Secondary | Hosewife | Farmer | 5 | 1 | Low |
| 4 | 24 | 29 | Secondary | Secondary | Hosewife | Driver | 6 | 0 | Low |
| 5 | 21 | 20 | High school diploma | High school diploma | Hosewife | Self-employed | 1 | 0 | Moderate |
| 6 | 40 | 50 | High school | High School diploma | Tailor | Bus driver | 23 | 3 | High |
| 7 | 21 | 31 | High school diploma | Master's degree | Hosewife | Self-employed | 5 | 0 | High |
| 8 | 21 | 25 | Secondary | Secondary | Hosewife | Plasterworker | 7 | 1 | Low |
| 9 | 28 | 37 | High school diploma | Elementary | Hosewife | Worker | 10 | 2 | Low |
| 10 | 32 | 32 | Bachelor's degree | Master's degree | Goldsmith | Financial manager | 1 | 0 | Moderate |
| 11 | 36 | 48 | Bachelor's degree | Doctorate degree | Educational coach | Military | 17 | 2 | Good |
| 12 | 22 | 23 | Bachelor's degree | Secondary | Student | Self-employed | 2 | 0 | Moderate |
| 13 | 33 | 49 | high school Diploma | High school diploma | Worker | Worker | 17 | 1 | Low |
| 14 | 38 | 43 | Elementary | Elementary | Hosewife | Worker | 25 | 4 | Low |
| 15 | 28 | 32 | Secondary | High school diploma | Hairdresser | Worker | 10 | 1 | Low |
| 16 | 15 | 26 | Secondary | Secondary | Student | Buyer and seller of scrap | 2 | 0 | Low |
| 17 | 36 | 37 | Elementary | High school diploma | Worker | Worker | 6 | 1 | Low |
| 18 | 42 | 50 | Elementary | Elementary | Hosewife | Hotel security | 12 | 1 | Moderate |
| 19 | 32 | 36 | Bachelor's degree | Master's degree | Midwife | Hospital staff | 7 | 1 | High |
| 20 | 36 | 53 | Master's degree | Master's degree | Dental assistant | Out of job | 3 | 0 | Low |

Therefore, the researcher continued sampling until data saturation was reached. Inclusion criteria for participation in the study included: married women aged 15–49 years, no prior participation in qualitative interviews on domestic violence, ability to speak Persian, literacy in reading and writing.

## Data collection

A semi -structured, in-depth interview was conducted with 20 women who had experienced domestic violence. Demographic information of the participants was obtained by the first author at the beginning of each interview. The participants' characteristics are presented in Table 1. The data were collected by the first author, who has experience in conducting qualitative studies and has 5 years of experience in the field. All interviews were conducted in a private room and at a convenient time for the participants to ensure the privacy and confidentiality of their information. After obtaining consent from the participants, the interviews were recorded and field notes were taken. Data saturation was achieved after 18 interviews. To ensure data saturation, 2 additional interviews were conducted, during which no new data emerged. On average, the interviews lasted for 60 minutes.

## Data analysis

To answer the research question, conventional content analysis was employed. Data were analyzed concurrently with data collection, following the approach proposed by Zhang and Wildemuth (2016) [22,23]. This approach allows researchers to examine individual experiences and indicate conflicting opinions and unresolved issues regarding the meaning and use of concepts, procedures, and interpretations through MAXQDA software (version 10, VERBI Software, Berlin, Germany) [24–27]. After each interview, the primary researcher listened to it multiple times to gain an overall understanding of the content. Then, the interview was transcribed word for word, and repeatedly read to achieve a deeper understanding of the data. The text of each interview was divided into meaningful units. Subsequently, these meaningful units were condensed, summarized, and coded. Codes were compared and categorized based on similarities and differences.

## Trustworthiness

To uphold the rigor and trustworthiness of the data, credibility, dependability, confirmability, and transferability were employed as guiding principles [24,28]. To ensure the credibility of the results, three expert reviewers (professors specializing in reproductive health) rigorously examined and confirmed the processes of data collection, data analysis, and interpretation of the results. For this purpose, purposive sampling with maximum variation and appropriate participants and key informants was utilized. To maintain dependability, the accuracy of data analysis was confirmed by three independent skilled researchers in qualitative research (assistant professors of reproductive health). Confirmability was addressed by providing some transcripts along with codes and categorizations to ZB and two other faculty members outside the research field with expertise in qualitative research, who confirmed the analysis process. Transferability was ensured by elaborating on study characteristics such as the research location, participants, and data collection and data analysis processes in detail for further evaluation.

## Results

The participants consisted of 20 women who were victims of domestic violence. The age range of the participating women was 15–45 years old, with an average age of 30.65 years. The educational level of the women ranged from elementary school to postgraduate degrees, and the educational backgrounds of their spouses varied from elementary school to doctorate degrees (Table 1).

The data provided insights into women's experiences of seeking formal and informal help, reflecting the pathways of seeking assistance, family support or lack thereof, the consequences of seeking help, inhibiting factors for seeking help, and factors contributing to domestic violence. Data analysis revealed 3 main categories and 12 sub-categories.

The first category was " structural and economic barriers to empowerment," the second category was " ineffective support providers," and the third category was " efforts and struggles to preserve the family," further details of the 200 results, including sub-categories, are presented in Table 2.

## Category 1: Structural and economic barriers to empowerment

One related barrier, as expressed by participants, is the fear of social consequences resulting from reporting spousal abuse to official channels. Particularly, women fear that reporting violence may lead to divorce or abandonment, leaving them without any financial support. Structural factors such as poverty and patriarchal inheritance patterns contribute to women's economic dependence on men, thus adding to the barriers to seeking help. Category 1 included five subcategories: A) High costs of legal and social support services, B) Lack of economic independence for women, C) Lack of awareness of one's rights, D) Lack of access to education and employment opportunities, and E) Belief in women's vulnerability after divorce

**A. High costs of legal and social support services.** One significant barrier to women victims of domestic violence seeking help is the high cost associated with legal action. Therefore, women victims of domestic violence with low financial capabilities require intersectoral, legal, social, and government financial support to pursue legal remedies. The following quote illustrates this:

*"I struggled a lot myself because legal complaints and divorce proceedings require a lot of money. My husband had hired a lawyer, and I couldn't afford to hire one myself" (Participant 12, 22 years old).*

**Table 2. Categories and subcategories.**

| Categories | Subcategories | Codes |
|---|---|---|
| 1. Structural and economic barriers to empowerment | A) High costs of legal and social support services | -Financial Challenges in Legal Processes<br>-Disparity in Access to Legal Representation |
| | B) Lack of economic independence for women | -Lack of Financial Independence as a Cause of Violence<br>-Financial Barriers to Seeking Justice<br>-Exploitation of Financial Dependence |
| | C) Lack of awareness of one's rights | -Lack of Education and Awareness in Families<br>-Need for Media and Educational Programs |
| | D) Lack of access to education and employment opportunities | -Self-Improvement and Empowerment<br>-Importance of Education and Critical Skills |
| | E) Belief in women's vulnerability after divorce | -Exploitation of Divorced Women<br>-Challenges in the Divorce Process |
| 2. Ineffective support providers | A) Ineffective support organizations | -Ineffectiveness of Emergency Social Services<br>-Limitations of Welfare Organization Support |
| | B) Concerns about the ineffectiveness of legal complaints | -Lack of Legal Consequences for Abusive Behavior<br>-Invisibility of Abuse |
| | C) Unsupportive Responses from Extended Family Members | - Lack of Family Support<br>- Pressure to Tolerate Abuse for the Sake of Family or Children |
| 3. Efforts and struggles to preserve the family | A) Fear of stigma | -Shame and Fear of Judgment<br>-Fear of Blame and Ridicule |
| | B) Fear of divorce | -Fear of Returning to Parental Home<br>-Fear of Becoming a Burden |
| | C) Concern for the future of their children | -Fear of Losing Custody<br>-Concern for the Child's Future |
| | D) Uncertain fate | -Sacrifice and Investment in Family Life<br>-Challenges of Starting Over |

**B. Lack of economic independence for women.** Reducing women's economic dependence on their spouses through entrepreneurship, skill-building, and income generation can lead to their job advancement and reduction of domestic violence. A female victim explains the importance of financial independence as follows:

*"One of the main reasons for violence is that the woman is not financially independent. A woman should never rely on a man. Now, I spent a lot of money just for a complaint. When I am oppressed, maybe I won't even have the money to come here. If I didn't have the 130,000 tomans they took from me for the legal examination, would they let me in? My husband knows I don't have financial support, and no matter what he does, I have to endure it"(Participant 9, 28 years old).*

**C. Lack of awareness of one's rights.** Increasing awareness among women and men about their individual rights, defining violence and instances of domestic violence, along with enhancing societal awareness in this regard, empower them and reduce violence. Gender inequality is the most prevalent and enduring factor affecting various aspects of women's lives. The patriarchal environment prevailing in families and women's lack of awareness of their rights are significant structural and cultural barriers to preventing women from seeking help. The following quotes reflect the importance of this issue:

*"We don't educate families at all; women are not empowered and are unaware of their rights. Besides my husband, my own father is also unaware of women's rights. I think television should have a dedicated channel for family formation and educating people about their rights, and there should be specific information for it." (Participant 10, 32 years old).*

*"My husband used to tell me he could do anything because he's a man. He said I had to be careful not to make any mistakes. They treated women like servants. My mother-in-law doesn't even eat at the same table as her husband and sons; poor thing, she's not even aware of her rights." (Participant 7, 21 years old).*

**D. Lack of access to education and employment opportunities.** There is a close relationship between low education levels among women and their exposure to domestic violence. Empowering women and improving their status in society through promoting gender equality in all rights, especially in education and employment, will be effective in reducing violence against them. A participant explained the importance of having higher education:

*"Women need to work on themselves in terms of education and behavior. They shouldn't take on a victim mentality and submit to injustice to avoid harm. When a woman lacks the power of speech, analytical skills, and higher education, she becomes vulnerable. In my opinion, the more empowered and informed a woman is, and the better she can articulate herself and analyze situations, the better she can stand up to such men."(Participant 19, 32 years old).*

**E. Belief in women's vulnerability after divorce.** Due to cultural and structural misconceptions about divorced women and their lack of financial independence after divorce, women are perceived as more vulnerable compared to men. Therefore, strengthening intersectoral support and receiving social support, as well as providing income-generating skills to women who are victims of violence, can be very effective. A participant explained:

*"Nowadays, men take advantage of divorced women. Maybe if a man who has never looked at me before realizes I'm divorced, he'll look at me differently to satisfy his desires. In my opinion, the law should consider some rights for women. For example, provide them with loans or offer them job opportunities. During divorce, the law and government should support women. Women are really suffering from the divorce process. I'm exhausted from all the back and forth, and I don't have the financial means to hire a lawyer."(Participant 15, 28 years old).*

## Category 2: Ineffective support providers

Even women who decide to seek help despite the existing challenges face other obstacles, many of which are structural barriers within the framework of norms and gender biases.

Discriminatory behavior by service providers undermines the trust of domestic violence victims in the responsiveness of the system. In the worst-case scenario, formal assistance may even harm the victims. Therefore, for the minority of women who effectively pass the first two stages of seeking help (i.e., identifying violence as a problem and deciding to seek help), direct or indirect experience with an unresponsive system can be another significant deterrent. Moreover, despite recognizing that IPV violates their rights and that they should seek redress, participants expressed disappointment over the lack of support from their families. Category 2 included 3 subcategories: A) Ineffective Support Organizations, B) Concerns about the ineffectiveness of Legal Complaints and C) Unsupportive Responses from Extended Family Members.

**A. Ineffective support organizations.** The development and adoption of national policies for the implementation of comprehensive services to support women and prevent gender-based violence have been a crucial turning point and can standardize management. The Welfare Organization is one of the most important support organizations that provide emergency social services, telephone counseling, and safe houses for women victims of domestic violence. Strengthening welfare services can provide more effective support services for women victims of violence and reduce their concerns about seeking help. A participant in this regard explained:

*"I called the emergency social services hotline (123) over 50 times, unfortunately, they don't respond!! They only provide telephone counseling and say it's a family dispute, and that's it. The Welfare Organization only shelters you until the legal process of the complaint is completed. The law should give us importance"(Participant 1, 31 years old).*

**B. Concerns about the ineffectiveness of legal complaints.** Social acceptance of violence leads to the normalization of male violence and the acceptance of violence by women, making women doubt the effectiveness of legal complaints. Gender norms legitimize male dominance and portray beating as a means of enforcing the man's masculine ideals. A participant expressed:

*"I have not received any rights or privileges from my husband; he says if I report him to the courts, nothing will happen to him legally. Many things cannot even be complained about and are done behind closed doors, and you cannot prove anything." (Participant 19, 32 years old).*

**C. Unsupportive responses from extended family members.** Various individual, interpersonal, and cultural factors play a significant role in the process of seeking help and determining whether women victims of domestic violence seek external support. Therefore, social and cultural dynamics in reinforcing women's agency and, consequently, family support for victims of domestic violence are important facilitators in seeking help. A participant who was upset about the lack of support from her family said:

*My husband used to beat and swear at me since the early days of our marriage. But my family didn't support me. They said we had no choice, we wanted you to stay home and keep your life. Once when my husband broke my tooth, I wanted to file a complaint, but my sister stopped me, she said it's close to Nowruz, don't do this, don't complain because of your children" (Participant 2, 48 years old).*

## Category 3: Efforts and struggles to preserve the family

For most women, seeking help is perceived as possible only when they decide to leave the relationship. Many of them prioritize their role as mothers over their own rights. They feel that their mission from marriage is to preserve the family,

and despite experiencing violence from their spouse, they are not willing to seek help or leave their spouse because they feel a great responsibility towards their children. They express that their children are the real victims of domestic violence. A wide range of structural, cultural, and social variables hinder both formal and informal help-seeking behaviors among women who are victims of domestic violence. Social-cultural barriers in a society are imposed by prevalent norms, values, and attitudes towards domestic violence victims. Structural barriers are determined by processes and legal frameworks. These hindrances lead to a reduction in women's help-seeking behaviors. Category 3 included four subcategories: A) Fear of Stigma, B) Fear of divorce, C) Concern for the future of their children, and D) Uncertain fate.

**A. Fear of stigma.** In the societal context of Iran, societal values outweigh individual issues for divorced women. Concerns about reputation, judgment, or losing social circles and society's negative view of divorced women hinder help-seeking behaviors among domestic violence victims. The following quote highlights the significance of the fear of stigma:

*"Their perception of a divorced person is negative. I feel embarrassed to talk to anyone about my husband's violence because I feel my human worth diminishes. I fear being blamed and facing merciless ridicule after divorce." (Participant 11, 36 years old).*

**B. Fear of divorce.** Marriage has potential benefits, and the thought of losing those benefits after divorce leads to fear among women. Additionally, the societal attitude in Iranian culture perceives a woman's identity within the framework of the family. Therefore, breaking this nucleus disrupts the woman's familial identity and personality. A participant who was deeply upset about this issue stated:

*"Every girl enters her husband's house with a thousand hopes and dreams. She wants her conditions in her husband's house to be better than her father's house and to once and for all relieve the burden from her family's shoulders. It's very bad for a girl who has left her father's house to go back there again. I'm very afraid of becoming a burden to my family after divorce because it will take a long time for me to become financially independent."(Participant 15, 32 years old).*

**C. Concern for the future of their children.** Motherhood is one of the most important roles a woman can play, so concern for the future of children is one of the most common fears for women. Maternal affection and facing the custody laws, along with the fear that custody arrangements may not be in their favor and that their children might be taken away from them, discourage many women from seeking help for domestic violence, because under Iranian custody laws, after divorce, custody is often awarded to the father or paternal relatives rather than the mother, which creates additional challenges for women. The following quote illustrates the significance of this issue:

*"I endured that life because of my child, and I didn't want to abandon my little one. I remained silent because my child was at an age where I feared that if I took action, they would take my child away from me and I was afraid for his future. If I had divorced when my son was young, he wouldn't have turned out so innocent like this." (Participant 13, 33 years old).*

**D. Uncertain fate.** Mental exhaustion and the realization that they may have to leave behind the home and life they painstakingly built and navigate many paths alone from scratch lead to women's fear of the uncertain fate they may face after divorce. A participant in this regard stated:

*"I spent my youth building that life for 23 years. Now, I have to gather my strength with three children. I don't know what will happen to me and my children after divorce, but I'll take the divorce, rent a house, and raise my children peacefully." (Participant 6, 40 years old).*

## Discussion

This is the first qualitative study in Iran specifically aimed to elucidate understanding and experiences of Iranian victims of domestic violence regarding barriers to seeking help. The present study demonstrates that women who are victims of domestic violence in Iran do not receive support services that establish justice and deserve it. This in turn leads the majority of domestic violence victims to refrain from seeking help to hold the perpetrator accountable, thus failing to prevent further violence. Low levels of assistance result from various structural barriers, such as social and cultural obstacles, which hinder victims from seeking help from both formal and informal sources. Among these factors, accepting IPV as a gender norm and instilling a sense of shame and embarrassment in help-seeking recipients are noteworthy. This leads victims of violence to prefer silence because they fear social consequences such as divorce, humiliation, and being ostracized by their families. In patriarchal societies, seeking help from domestic violence victims seems to result in the shame of those around them rather than the violent act itself. The cultural and social environment of Iran significantly influences women's experiences of domestic violence and their capacity to seek assistance. A more subtle cause is the patriarchal values embedded in our society which not only bolster male supremacy but also discourage women from reporting domestic abuse. Many women fear social stigma, family shame or a potential backlash from their abuser if they seek external help. Furthermore, women's rights and protection are restricted by legal and religious systems. Iran's judicial system, informed by Islamic law, often favors men in family and legal disputes. For instance, a woman seeking a divorce due to domestic violence, must provide substantial evidence, which is often difficult to obtain such as medical certificates, police reports, or witness testimonies and the methods for doing so are still extremely harsh. Other regulations, such as child custody and economic assistance laws, further discourage women from leaving abusive relations out of fear that they will lose their children or economic security. The other major factor is the stigma surrounding divorce, which has a significant impact on women's decisions to reach out for help. Divorced women often face social ostracism, decreased chances of remarriage and other opportunities, and impoverishment, which can create a reluctance to leave abusive marriages. Pressure to "protect the family" often supersedes personal health, compelling many women to remain in toxic situations.

The findings of the current study are consistent with the results of studies conducted by McCleary-Sills and colleagues (2016) and Jakobsen (2014), where gender norms legitimize IPV and portray women as obedient individuals who should obey their husbands as heads of the family even at the expense of sacrificing their own rights [29,30]. Victims of domestic violence often experience powerful barriers to disclosing the violence, which leads to learned helplessness and subsequently jeopardizes their mental health and physical safety. Therefore, most of them perceive violence as normal and consequently are deprived of taking the first step in the help-seeking process. The results of a study by Choden (2019) indicated that even women who recognize IPV as a violation of their rights are unlikely to progress in the help-seeking process because they lack sufficient social support to report violence through official channels. Consequently, women who successfully overcome these initial barriers prefer silence due to mistrust that service providers will adequately and fairly meet their needs [10]. These findings are consistent with the results of the present study.

Most participants in the present study expressed a lack of necessary and sufficient information about their rights and noted insufficient education in this regard. These findings are consistent with the results of studies conducted by Deuba *et al*. (2016) and Sabri *et al*. (2015), which emphasized the necessity of educating women to reduce violence and developing awareness programs in this area. Their studies showed that couples should receive education on reproductive health, such as safe sexual life, understanding the roles of women and men, maintaining individual values such as women's rights, self-esteem, and women's self-awareness, and be informed about the adverse effects of violence on the health of women and children, and empower themselves in this regard [31,32].

Most victims of violence expressed that one of the main barriers to seeking help and one of the significant factors contributing to women's exposure to violence is their lack of financial independence and economic dependence on their spouses. In Iran, many employers tend to prefer hiring men, and under equal conditions, women have fewer opportunities for employment. Additionally, in some regions of Iran, men prevent their wives and daughters from pursuing employment

or continuing their education, which poses a significant barrier to women's financial independence. Although women who experience violence often face serious economic hardship and may expect support from government institutions, health-care systems, or social workers, in reality, such support is often scarce, inaccessible, or insufficient. This finding is consistent with the results of a study by Fogarty et al. (2019), which emphasized the importance of financial support for women victims of IPV [33].

Victims of domestic violence in Iran lack sufficient support from the judicial system, so the legal system needs reform to provide effective services to women victims of violence. They need legal assistance in child custody cases, which often forces most women to endure a life of violence against their will and in violation of their rights. This finding is consistent with the results of a study by Sigalla et al. (2018), which supported considering legal aid for child custody [34].

The results of the present study indicate that women victims of violence, especially when faced with severe physical violence, need assistance from the police and the criminal justice system. Therefore, it is essential to assess and reform the protocols of these support systems according to the needs of women victims [35]. These findings are consistent with the results of a study by Gashaw et al. (2019), in which police officers stated that they prefer marital reconciliation over women defending their rights, which may lead to divorce [36]. The authors suggest these recommendations tailored to the Iranian context to support women against domestic violence:

- Creating community support centers for domestic violence victims: Establishing community support centers in Ardabil province that provide the necessary legal, psychological, and job training for women affected by domestic violence. They may be set up in conjunction with community health care centers to make it more convenient for people. Concerning the enhancement of current support systems, we suggest bolstering and broadening the functions of welfare organizations, shelters, and law enforcement agencies. In particular, we suggest that these organizations obtain increased financial support, specialized training regarding domestic violence, and enhanced legal protection to safeguard victims. A more organized strategy for coordination among the police, social services, and healthcare providers is crucial to guarantee that women receive the assistance they require at all phases

- Economic Empowerment Programs: Develop government-funded programs that provide women with access to micro-loans or grants to start microenterprises, thus reducing their economic dependence on their abuser.

- Cultural Awareness Initiatives: Work hand in hand with media campaigns and community and local faith leaders to culturally interrogate patriarchal norms and decrease the stigma around asking for help for domestic violence.

- Law Enforcement Practices: Ensure that police departments and the judicial system receive mandatory training and standards for responding to domestic violence and treatment of victims with respect and dignity.

## Limitations

This study was conducted on a limited number of female victims of domestic violence in one city in Iran; therefore, the results could not be generalized to all Iranian victims of domestic violence. Furthermore, translation of the interviews from Turkish to Farsi and English were a limitation of the study that were performed separately by two proficient translators and were then checked by a third person.

## Conclusion

Data analyses reveal the persistence of patriarchal norms in Iranian society and the systematic violations of legal rights, which constitutes the initial barrier for women when confronting domestic violence. This research shows that women find access to essential support services unattainable due to financial dependence on the abuser, oppressive legal frameworks, and the fear of social stigma. Moreover, both the legal system and social support structures fail them effectively, often leaving them trapped in abusive situations.

The high costs of legal processes and the long journey required to get satisfactory results from existing support organizations are significant challenges according to key findings. Economic and legal barriers prevent many women from reporting abuse or leaving their abusers. The stigma attached to divorce and the societal expectations of upholding family unity is another reason that prevents women from seeking help.

The study notes that such reforms will not be simple, but they are attainable. Iranian women can escape from domestic violence if such support systems are in place along with a supportive group of legal practitioners.

## Author contributions

**Conceptualization:** Samaneh Dabagh Fekri.

**Data curation:** Samaneh Dabagh Fekri, Negar Khoshnevis, Elham Kheirkhahi.

**Formal analysis:** Armin Zareiyan.

**Methodology:** Masoumeh Namazi.

**Supervision:** Zahra Behboodi Moghadam.

**Writing – original draft:** Samaneh Dabagh Fekri.

**Writing – review & editing:** Samaneh Dabagh Fekri.

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
