## [Decision Letter · Decision Letter 0]

PONE-D-24-60271Breaking the Silence: Barriers to Help-Seeking among Female Victims of Domestic Violence – A Qualitative StudyPLOS ONE

Dear Dr.  Behboodi Moghadam

Thank you for submitting your manuscript to PLOS ONE. After careful consideration, we feel that it has merit but does not fully meet PLOS ONE’s publication criteria as it currently stands. Therefore, we invite you to submit a revised version of the manuscript that addresses the points raised during the review process.

Please note that I have acted as a reviewer for this manuscript, and you will find my comments below, under Reviewer 1.

We look forward to receiving your revised manuscript.

Kind regards,

Shadab Shahali, PHD

Academic Editor

PLOS ONE

Journal Requirements:

Reviewers' comments:

Reviewer's Responses to Questions

**Comments to the Author**

1. Is the manuscript technically sound, and do the data support the conclusions?

Reviewer #1: Partly

Reviewer #2: Yes

2. Has the statistical analysis been performed appropriately and rigorously? 

Reviewer #1: N/A

Reviewer #2: Yes

3. Have the authors made all data underlying the findings in their manuscript fully available?

Reviewer #1: No

Reviewer #2: Yes

4. Is the manuscript presented in an intelligible fashion and written in standard English?

Reviewer #1: No

Reviewer #2: Yes

5. Review Comments to the Author

Reviewer #1: Title:

Add a geographical or cultural indicator in the title to emphasize the study's regional and cultural relevance. For example, explicitly mentioning Iran or Ardabil province will highlight the contextual specificity of the research.

Innovative Contributions:

The study briefly discusses the necessity of "specialized services" and "financial and moral support" but does not propose innovative or unique solutions. To enhance its contribution, the authors should explore novel recommendations tailored to the Iranian context, particularly focusing on actionable strategies for systemic reform.

Results Section:

While the results are well-structured into categories and subcategories, the explanations of these themes lack depth. Certain categories, such as "Unfavorable Socioeconomic Status" and "Lack of Women's Empowerment," overlap conceptually, which dilutes their distinctiveness. Consider merging overlapping categories or providing clearer distinctions.

Some subcategories are underdeveloped (e.g., "Lack of Economic Independence for Women"). These should be critically analyzed in more detail, with stronger contextualization in terms of societal and policy implications.

The participant voices included are valuable, but the range of perspectives is limited. Ensure that diverse viewpoints, such as those of women from different age groups, education levels, and urban/rural settings, are adequately represented.

Novelty and Contextual Focus:

The results do not highlight particularly unique or novel insights that distinguish this study from similar research in other settings. Emphasize findings that are specific to the cultural and social context of Iran to underscore the study's uniqueness.

In both the results and discussion, ensure that claims about societal and structural barriers are explicitly linked back to the study's cultural and geographical context.

Discussion:

The discussion makes broad claims without adequately emphasizing the study's specific contributions to the field of domestic violence research. Highlight how this study advances knowledge, particularly within reproductive health and gender-based violence in patriarchal societies.

While the authors propose general solutions (e.g., breaking cultural taboos, raising awareness), actionable and specific recommendations are lacking. For instance:

What specific policy reforms or legal changes are necessary in the Iranian context?

How can existing support systems (e.g., police, welfare organizations) be improved to address the barriers identified in the study?

Including a comparison with findings from other regions or countries with similar patriarchal norms could further enrich the discussion and provide valuable context.

Conclusion:

The conclusion reiterates points already made in the discussion and lacks a strong final message. Make it more concise and focused on the study's key takeaways, highlighting the most critical findings, their implications for reproductive health, and clear recommendations for policy and practice.

Reviewer #2: The article is very significant and valuable and needs minor corrections.

Abstract

1-Please write more important findings about age and education level in the results section.

Background:

1-Update all statistics provided in the background. 1993 is too old.

2-The introduction is very long and lacks a clear procedure.

3-What is your novelty? Gap knowledge?

4-Please summarize the study aim and add it to the last section of the background.

Methods:

1-Lines 143: Maximum variation is correct

2-lines 144: Cite Table 1

Results:

1-Lines 196: First mention main categories and then sub-categories

2-In Table 2: Point out and provide some codes in Table 2 or the text.

6. PLOS authors have the option to publish the peer review history of their article (what does this mean? ). If published, this will include your full peer review and any attached files.

**Do you want your identity to be public for this peer review?** For information about this choice, including consent withdrawal, please see our Privacy Policy .

Reviewer #1: No

Reviewer #2: No

---

## [Author Response · Author response to Decision Letter 1]

12 Apr 2025

Dear Editor,

We appreciate you and the reviewers for your precious time in reviewing our paper and providing valuable comments. It was your valuable and insightful comments that led to possible improvements in the current version. The authors have carefully considered the comments and tried our best to address every one of them. We hope the manuscript after careful revisions meet your high standards. The authors welcome further constructive comments if any.

Below we provide the point-by-point responses. All modifications in the manuscript have been highlighted in yellow.

Sincerely,

Zahra Behboodi Moghadam

Professor, School of Nursing & Midwifery, Tehran University of Medical Sciences, Tehran, Iran.

e-mail: Behboodi@tums.ac.ir

Reviewer1:

Comment 1. Title: Add a geographical or cultural indicator in the title to emphasize the study's regional and cultural relevance. For example, explicitly mentioning Iran or Ardabil province will highlight the contextual specificity of the research.

• Reply 1: Thanks for your kind reminders. This is corrected and highlighted in the manuscripts.

• Revised Title: "Breaking the Silence: Barriers to Help-Seeking among Female Victims of Domestic Violence in Ardabil, Iran – A Qualitative Study"

Comment 2. The study briefly discusses the necessity of "specialized services" and "financial and moral support" but does not propose innovative or unique solutions. To enhance its contribution, the authors should explore novel recommendations tailored to the Iranian context, particularly focusing on actionable strategies for systemic reform.

• Reply 2: Thanks for your kind reminders. This is corrected and highlighted in the manuscripts.

Contextual suggestions for problem-solving are also provided in discussion section:

-Creating community support centers for domestic violence victims: Establishing community support centers in Ardabil province that provide the necessary legal, psychological, and job training for women affected by domestic violence. They may be set up in conjunction with community health care centers to make it more convenient for people.

-Economic Empowerment Programs: Develop government-funded programs that provide women with access to microloans or grants to start microenterprises, thus reducing their economic dependence on their abuser.

-Cultural Awareness Initiatives: Work hand in hand with media campaigns and community and local faith leaders to culturally interrogate patriarchal norms and decrease the stigma around asking for help for domestic violence.

-Law Enforcement Practices: Ensure that police departments and the judicial system receive mandatory training and standards for responding to domestic violence and treatment of victims with respect and dignity

Comment 3. Results Section: While the results are well-structured into categories and subcategories, the explanations of these themes lack depth. Certain categories, such as "Unfavourable Socioeconomic Status" and "Lack of Women's Empowerment," overlap conceptually, which dilutes their distinctiveness. Consider merging overlapping categories or providing clearer distinctions.

• Reply 3: Thanks for your kind reminders. This is corrected and highlighted in the manuscripts.

The categories "Unfavourable Socioeconomic Status" and "Lack of Women's Empowerment" merged into a single category named “Structural and Economic Barriers to Empowerment” with subcategories, including:

A) High costs of legal and social support services

B) Lack of economic independence for women

C) Lack of awareness of one’s rights

D) Lack of access to education and employment opportunities

E) Belief in women's vulnerability after divorce

Comment 4. Some subcategories are underdeveloped (e.g., "Lack of Economic Independence for Women"). These should be critically analyzed in more detail, with stronger contextualization in terms of societal and policy implications.

• Reply 4: Thanks for your kind reminders. This is corrected and highlighted in the manuscripts.

In Iran, many employers tend to prefer hiring men, and under equal conditions, women have fewer opportunities for employment. Additionally, in some regions of Iran, men prevent their wives and daughters from pursuing employment or continuing their education, which poses a significant barrier to women's financial independence.

Comment 5. The participant voices included are valuable, but the range of perspectives is limited. Ensure that diverse viewpoints, such as those of women from different age groups, education levels, and urban/rural settings, are adequately represented.

• Reply 5: Thanks for your kind reminders. In this study, purposive sampling with maximum variation was conducted, aiming to select women from diverse groups with different age ranges, educational backgrounds, and other characteristics.

Participants from diverse age groups (e.g., younger and older women), education levels (e.g., women with no formal education and those with advanced degrees), and urban/rural settings. This will provide a more comprehensive understanding of the barriers faced by women across different contexts.

o Younger women may face unique challenges related to societal expectations and lack of life experience.

o Women from rural areas may encounter additional barriers due to limited access to support services.

• Women with higher education levels may still face cultural stigma despite their educational attainment.

Comment 6. Novelty and Contextual Focus: The results do not highlight particularly unique or novel insights that distinguish this study from similar research in other settings. Emphasize findings that are specific to the cultural and social context of Iran to underscore the study's uniqueness. In both the results and discussion, ensure that claims about societal and structural barriers are explicitly linked back to the study's cultural and geographical context.

• Reply 6: Thanks for your kind reminders. This is corrected and highlighted in the manuscripts.

The cultural and social environment of Iran significantly influences women's experiences of domestic violence and their capacity to seek assistance. A more subtle cause is the patriarchal values embedded in our society which not only bolster male supremacy but also discourage women from reporting domestic abuse. Many women fear social stigma, family shame or a potential backlash from their abuser if they seek external help. Furthermore, women's rights and protection are restricted by legal and religious systems. Iran’s judicial system, informed by Islamic law, often favors men in family and legal disputes. For instance, a woman seeking a divorce due to domestic violence must provide substantial evidence, which is often difficult to obtain — and the methods for doing so are still extremely harsh. Other regulations, such as child custody and economic assistance laws, further discourage women from leaving abusive relations out of fear that they will lose their children or economic security. The other major factor is the stigma surrounding divorce, which has a significant impact on women's decisions to reach out for help. Divorced women often face social ostracism, decreased chances of remarriage and other opportunities, and impoverishment, which can create a reluctance to leave abusive marriages. Pressure to “protect the family” often supersedes personal health, compelling many women to remain in toxic situations.

Comment 7. The discussion makes broad claims without adequately emphasizing the study's specific contributions to the field of domestic violence research. Highlight how this study advances knowledge, particularly within reproductive health and gender-based violence in patriarchal societies.

While the authors propose general solutions (e.g., breaking cultural taboos, raising awareness), actionable and specific recommendations are lacking. For instance:

What specific policy reforms or legal changes are necessary in the Iranian context?

How can existing support systems (e.g., police, welfare organizations) be improved to address the barriers identified in the study?

Including a comparison with findings from other regions or countries with similar patriarchal norms could further enrich the discussion and provide valuable context.

Reply 7: Thanks for your kind reminders. This is corrected and highlighted in the manuscripts.

- Progress in Understanding Regarding Domestic Violence and Reproductive Health : We acknowledge the importance of demonstrating how this research contributes to the joint body of knowledge on domestic violence — particularly regarding reproductive health issues and gender-based violence in patriarchal societies. This research offers unique perspectives on how domestic violence is perpetuated in Iran by patriarchal standards, legal systems, and socio-cultural barriers. Focusing on women in this specific social and cultural context, we highlight the impact that legal and religious frameworks have on women's autonomy, access to a support system, and reproductive rights. Our findings add to the understanding of how social expectations, legal barriers, and economic dependence affect reproductive health and gender-based violence, issues that are not as well-studied in similar research.

The authors suggest these recommendations tailored to the Iranian context to support women against domestic violence:

- Creating community support centers for domestic violence victims: Establishing community support centers in Ardabil province that provide the necessary legal, psychological, and job training for women affected by domestic violence. They may be set up in conjunction with community health care centers to make it more convenient for people. Concerning the enhancement of current support systems, we suggest bolstering and broadening the functions of welfare organizations, shelters, and law enforcement agencies. In particular, we suggest that these organizations obtain increased financial support, specialized training regarding domestic violence, and enhanced legal protection to safeguard victims. A more organized strategy for coordination among the police, social services, and healthcare providers is crucial to guarantee that women receive the assistance they require at all phases

- Economic Empowerment Programs: Develop government-funded programs that provide women with access to microloans or grants to start microenterprises, thus reducing their economic dependence on their abuser.

- Cultural Awareness Initiatives: Work hand in hand with media campaigns and community and local faith leaders to culturally interrogate patriarchal norms and decrease the stigma around asking for help for domestic violence.

- Law Enforcement Practices: Ensure that police departments and the judicial system receive mandatory training and standards for responding to domestic violence and treatment of victims with respect and dignity

Comment 8. Conclusion: The conclusion reiterates points already made in the discussion and lacks a strong final message. Make it more concise and focused on the study's key takeaways, highlighting the most critical findings, their implications for reproductive health, and clear recommendations for policy and practice.

Reply 8: Thanks for your kind reminders. This is corrected and highlighted in the manuscripts.

Data analyses reveal the persistence of patriarchal norms in Iranian society and the systematic violations of legal rights, which constitutes the initial barrier for women when confronting domestic violence. This research shows that women find access to essential support services unattainable due to financial dependence on the abuser, oppressive legal frameworks, and the fear of social stigma. Moreover, both the legal system and social support structures fail them effectively, often leaving them trapped in abusive situations.

The high costs of legal processes and the long journey required to get satisfactory results from existing support organizations are significant challenges according to key findings. Economic and legal barriers prevent many women from reporting abuse or leaving their abusers. The stigma attached to divorce and the societal expectations of upholding family unity is another reason that prevents women from seeking help.

The study notes that such reforms will not be simple, but they are attainable. Iranian women can escape from domestic violence if such support systems are in place along with a supportive group of legal practitioners.

Reviewer 2:

The article is very significant and valuable and needs minor corrections.

Reply: thank you very much.

Comment 1. Abstract : Please write more important findings about age and education level in the results section.

Reply 1. Thanks for your kind reminders. This is corrected and highlighted in the manuscripts.

Background:

comment 2. Update all statistics provided in the background. 1993 is too old.

Reply 2. Thanks for your kind reminders. This is corrected and highlighted in the manuscripts.

Update Statistics: Globally, 1 in 3 women (approximately 30%) experience physical or sexual violence in their lifetime, with intimate partner violence being the most common form (WHO, 2021). In Iran, recent studies indicate that domestic violence affects up to 66% of women, with significant regional variations (Alamolhoda et al., 2024).

comment 3. The introduction is very long and lacks a clear procedure.

Reply 3. Thanks for your kind reminders. This is summarized.

comment 4. What is your novelty? Gap knowledge?

Reply 4. Thanks for your kind reminders. This is corrected and highlighted in the manuscripts.

While numerous studies have explored domestic violence globally, there is limited qualitative research on the barriers to help-seeking among women in Iran, particularly in regions like Ardabil province. This study addresses this gap by providing an in-depth understanding of the cultural, social, and structural barriers faced by Iranian women, offering context-specific insights that are crucial for developing effective interventions.

comment 5. Please summarize the study aim and add it to the last section of the background.

Reply 5. Thanks for your kind reminders. This is corrected and highlighted in the manuscripts.

This study aims to explore the barriers to help-seeking among female victims of domestic violence in Ardabil, Iran, with a focus on understanding the cultural, social, and structural factors that influence their decisions.

Methods:

comment 6. Lines 143: Maximum variation is correct

Reply 6. Thanks for your kind reminders. This is corrected and highlighted in the manuscripts.

comment 7. lines 144: Cite Table 1

Reply 7. Thanks for your kind reminders. This is corrected and highlighted in the manuscripts.

Results:

comment 8. Lines 196: First mention main categories and then sub-categories

Reply 8. Thanks for your kind reminders. This is corrected and highlighted in the manuscripts.

comment 9. In Table 2: Point out and provide some codes in Table 2 or the text.

Reply 9. Thanks for your kind reminders. We added some codes in Table 2.

Dear Chief Editor,

We would like to thank you and the reviewers for the detailed feedback and for considering our manuscript for publication at PLOS ONE. We have adequately responded to all the requested changes, as detailed below:

Duplicates: We checked the submitted files and have excluded all unnecessary or duplicate files so only the relevant files for the new version of the manuscript are included.

Consistency title: The title has been checked to be consistent between the online submission form and the manuscript.

Data Availability Statement: We acknowledge that qualitative interview data are "data," according to PLOS ONE.

(a) Due to ethical reasons, the full dataset cannot be shared as it contains potentially identifying and sensitive information about participants. The limitations are enforced by Ethics Committee of Tehran University of Medical Sciences (ethics code: IR.TUMS.FNM.REC.1402.094).

(b) The data access policy may be directed to Ethics Committee of Tehran University of Medical Sciences.

Obtaining consen

---

## [Decision Letter · Decision Letter 1]

PONE-D-24-60271R1Breaking the Silence: Barriers to Help-Seeking among Female Victims of

 Domestic Violence in Ardabil, Iran – A Qualitative StudyPLOS ONE

Dear Dr. Behboodi Moghadam,

Thank you for submitting your manuscript to PLOS ONE. After careful consideration, we feel that it has merit but does not fully meet PLOS ONE’s publication criteria as it currently stands. Therefore, we invite you to submit a revised version of the manuscript that addresses the points raised during the review process.

We look forward to receiving your revised manuscript.

Kind regards,

Shadab Shahali, PHD

Academic Editor

PLOS ONE

Journal Requirements:

Reviewers' comments:

Reviewer's Responses to Questions

**Comments to the Author**

1. If the authors have adequately addressed your comments raised in a previous round of review and you feel that this manuscript is now acceptable for publication, you may indicate that here to bypass the “Comments to the Author” section, enter your conflict of interest statement in the “Confidential to Editor” section, and submit your "Accept" recommendation.

Reviewer #2: All comments have been addressed

Reviewer #3: (No Response)

2. Is the manuscript technically sound, and do the data support the conclusions?

Reviewer #2: Yes

Reviewer #3: Yes

3. Has the statistical analysis been performed appropriately and rigorously? 

Reviewer #2: Yes

Reviewer #3: Yes

4. Have the authors made all data underlying the findings in their manuscript fully available?

Reviewer #2: Yes

Reviewer #3: Yes

5. Is the manuscript presented in an intelligible fashion and written in standard English?

Reviewer #2: Yes

Reviewer #3: Yes

6. Review Comments to the Author

Reviewer #2: Thank you

you have been addressed all concerns.............................................................

Reviewer #3: This is my first review of this manuscript; I did not review the original submission.

This is an important study that will undoubtedly add to knowledge about Iranian women’s experiences of IPV. Methods of data collection and analysis are appropriate and well-described.

Line 82: Is “preferred silence” correct, according to the cited study, or is it that these women did not seek help? (Perhaps because of fear or a lack of avenues to do so.)

Line 93: I tend to think that the term “exposed” sounds like it could be witnessing; “experiencing” clarifies that they were direct victims of violence.

97- 98: unhelpful responses from service providers?

Results:

Table 2 provides a useful overview of the data (categories, subcategories, and codes).

Excellent inclusion of relevant quotes.

Recommendations (such as information, education, and opportunities for financial independence) are directly tied to the study’s findings.

I find the subtitle of “Unprincipled Family Support” to be a bit unclear—perhaps “Lack of Family Support” or “Unsupportive Responses from Extended Family Members?”

Lines 335- 338 could be clarified (regarding custody laws and how care of children after divorce may not be awarded to women)

Discussion:

Excellent points regarding patriarchal societies and the imposition shame and stigma on victims rather than on those who use violence.

Important points regarding Iran’s judicial system and Islamic law.

Regarding line 372 “must provide substantial evidence, which is often difficult to obtain — and the methods for doing so are still extremely harsh,” could some of the method be provided with e.g. in brackets?

Regarding line 409, “Women who are victims of violence face economic problems and expect the government, healthcare systems, or social workers to help them in this regard.” Is this a realistic expectation? Are these services and systems available to help victims of IPV with financial/economic challenges?

Excellent inclusion of recommendations for the Iranian context.

While I appreciate that translation and understanding women’s experiences when translated to English from their original language could be a limitation, it is a strength of this study that it makes this information available to readers in English.

Thank you to the authors for their work on this important topic.

7. PLOS authors have the option to publish the peer review history of their article (what does this mean? ). If published, this will include your full peer review and any attached files.

**Do you want your identity to be public for this peer review?** For information about this choice, including consent withdrawal, please see our Privacy Policy .

Reviewer #2: No

Reviewer #3: No

---

## [Author Response · Author response to Decision Letter 2]

2 Jun 2025

Dear Editor,

We appreciate you and the reviewers for your precious time in reviewing our paper and providing valuable comments. It was your valuable and insightful comments that led to possible improvements in the current version. The authors have carefully considered the comments and tried our best to address every one of them. We hope the manuscript after careful revisions meet your high standards. The authors welcome further constructive comments if any.

Below we provide the point-by-point responses. All modifications in the manuscript have been highlighted in yellow.

Sincerely,

Zahra Behboodi Moghadam

Professor, School of Nursing & Midwifery, Tehran University of Medical Sciences, Tehran, Iran.

e-mail: Behboodi@tums.ac.ir

Reviewer Comment 1:

This is an important study that will undoubtedly add to knowledge about Iranian women’s experiences of IPV. Methods of data collection and analysis are appropriate and well-described.

Response:

Thank you very much for your positive and encouraging feedback. we sincerely appreciate your kind words and are glad to hear that you found the study valuable and the methods well-described. Your support is truly appreciated.

Reviewer Comment 2:

Is “preferred silence” correct, according to the cited study, or is it that these women did not seek help? (Perhaps because of fear or a lack of avenues to do so.)

Response:

Thank you for your insightful comment. You are correct in noting the distinction. The phrase “Preferred silence” may unintentionally imply a conscious or voluntary choice, whereas the cited study suggests that the women did not seek help often due to fear, lack of support systems, or societal pressures. To reflect this more accurately, we have revised the text to clarify that their silence was not necessarily a preference but rather a consequence of limited options and external constraints.

(See Page 5, Line 81)

Reviewer Comment 3:

I tend to think that the term “exposed” sounds like it could be witnessing; “experiencing” clarifies that they were direct victims of violence.

Response:

Thank you for your insightful comment. We agree that the term "experiencing" is more accurate in this context, as it emphasizes the participants' direct victimization rather than passive observation. We have replaced “exposed” with “experiencing” throughout the manuscript.

(See Page 6, Line 94)

Reviewer Comment 4:

unhelpful responses from service providers?

Response:

Thank you for pointing this out. We have clarified this sentence to specify the types of unhelpful responses reported by participants. The revised version reads:

“Staying with an abusive partner is sometimes facilitated by unhelpful responses from service providers, such as being dismissed, blamed, or advised to remain silent for the sake of family unity.”

(See Page 6, Line99)

Results:

Reviewer Comment 5:

Provides a useful overview of the data (categories, subcategories, and codes). Excellent inclusion of relevant quotes. Recommendations (such as information, education, and opportunities for financial independence) are directly tied to the study’s findings.

Response:

Thank you for your positive and encouraging feedback. We appreciate your recognition of the clarity in Table 2 and the relevance of the quotes included. We are also glad that the linkage between the study’s findings and the recommendations was clear and meaningful. Your comments are greatly motivating and reinforce the direction of this research.

Reviewer Comment 6:

find the subtitle of “Unprincipled Family Support” to be a bit unclear—perhaps “Lack of Family Support” or “Unsupportive Responses from Extended Family Members?”

Response:

Thank you for this helpful suggestion. We agree that “Unprincipled Family Support” may be unclear or misleading. We have revised the subtitle to “Unsupportive Responses from Extended Family Members” throughout the manuscript to better reflect the intended meaning.

(See Page 13, 17, 18, Line 271, 292)

Reviewer Comment 7:

Lines 335-338 could be clarified (regarding custody laws and how care of children after divorce may not be awarded to women).

Response:

Thank you for pointing this out. We have revised the relevant section to provide clearer information about custody laws in Iran, highlighting that after divorce, “custody laws, along with the fear that custody arrangements may not be in their favor and that their children might be taken away from them, discourage many women from seeking help for domestic violence, because under Iranian custody laws, after divorce, custody is often awarded to the father or paternal relatives rather than the mother, which creates additional challenges for women.”

(See Page 21, Line 338)

Discussion:

Reviewer Comment 8:

Excellent points regarding patriarchal societies and the imposition shame and stigma on victims rather than on those who use violence.

Important points regarding Iran’s judicial system and Islamic law.

Response:

Thank you for your thoughtful comments on the discussion of patriarchal structures and the legal context in Iran. We are pleased that the analysis of the social stigma surrounding victims and the critical examination of the judicial and religious frameworks were found to be meaningful. These aspects were central to the study, and your recognition of their importance is truly appreciated.

Reviewer Comment 9:

Regarding line 372: “must provide substantial evidence, which is often difficult to obtain — and the methods for doing so are still extremely harsh,” could some of the method be provided with e.g. in brackets?

Response:

Thank you for your suggestion. We have revised the sentence to include examples of such methods. The updated sentence reads:

“A woman seeking a divorce due to domestic violence, must provide substantial evidence, which is often difficult to obtain such as medical certificates, police reports, or witness testimonies and the methods for doing so are still extremely harsh.”

(See Page 22, Line374)

Reviewer Comment 10:

Regarding line 409: “Women who are victims of violence face economic problems and expect the government, healthcare systems, or social workers to help them in this regard.” Is this a realistic expectation? Are these services and systems available to help victims of IPV with financial/economic challenges?

Response:

Thank you for this important comment. We recognize that although such expectations are common among victims, support systems in Iran, particularly in smaller cities like Ardabil, are often limited or unavailable. We have revised the sentence to reflect this reality more accurately:

“Although women who experience violence often face serious economic hardship and may expect support from government institutions, healthcare systems, or social workers, in reality, such support is often scarce, inaccessible, or insufficient.”

(See Page 24, Line412)

Reviewer Comment 11:

While I appreciate that translation and understanding women’s experiences when translated to English from their original language could be a limitation, it is a strength of this study that it makes this information available to readers in English.

Thank you to the authors for their work on this important topic.

Response:

Thank you for highlighting the value of contextualized recommendations and the effort to make participants’ voices accessible in English. We truly appreciate your recognition of the challenges and importance of translation in qualitative research. It was our intention to preserve the authenticity of women’s experiences while making them understandable and meaningful to an international audience.

We appreciate the constructive feedback and the opportunity to improve our manuscript. We believe the revisions have strengthened the clarity and rigor of our study, and we thank the reviewers and editor again for their time and expertise.

---

## [Decision Letter · Decision Letter 2]

Breaking the Silence: Barriers to Help-Seeking among Female Victims of

 Domestic Violence in Ardabil, Iran – A Qualitative Study

PONE-D-24-60271R2

Dear Dr. Behboodi Moghadam,

We’re pleased to inform you that your manuscript has been judged scientifically suitable for publication and will be formally accepted for publication once it meets all outstanding technical requirements.

Kind regards,

Shadab Shahali, PHD

Academic Editor

PLOS ONE

Additional Editor Comments (optional):

Reviewers' comments:

Reviewer's Responses to Questions

**Comments to the Author**

1. If the authors have adequately addressed your comments raised in a previous round of review and you feel that this manuscript is now acceptable for publication, you may indicate that here to bypass the “Comments to the Author” section, enter your conflict of interest statement in the “Confidential to Editor” section, and submit your "Accept" recommendation.

Reviewer #3: All comments have been addressed

2. Is the manuscript technically sound, and do the data support the conclusions?

Reviewer #3: Yes

3. Has the statistical analysis been performed appropriately and rigorously? 

Reviewer #3: N/A

4. Have the authors made all data underlying the findings in their manuscript fully available?

Reviewer #3: Yes

5. Is the manuscript presented in an intelligible fashion and written in standard English?

Reviewer #3: Yes

6. Review Comments to the Author

Reviewer #3: This is my second review of this manuscript.

The authors have addressed all of my previous comments, and I feel that the edits increase the clarity of the article. I appreciated how the authors highlighted the changes and corresponding reviewer comments.

This is an important study that will undoubtedly add to knowledge about Iranian women’s experiences of IPV.

I feel it is ready for publication.

7. PLOS authors have the option to publish the peer review history of their article (what does this mean? ). If published, this will include your full peer review and any attached files.

**Do you want your identity to be public for this peer review?** For information about this choice, including consent withdrawal, please see our Privacy Policy .

Reviewer #3: No

---

## [Editor Report · Acceptance letter]

PONE-D-24-60271R2

PLOS ONE

Dear Dr. Behboodi Moghadam,

I'm pleased to inform you that your manuscript has been deemed suitable for publication in PLOS ONE. Congratulations! Your manuscript is now being handed over to our production team.

Kind regards,

on behalf of

Dr. Shadab Shahali

Academic Editor

PLOS ONE